# Gaussian Process Conjoint Analysis for Adaptive Marginal Effect Estimation

**Yehu Chen, Jacob Montgomery, Roman Garnett**
Washington University in St Louis
`chenyehu,jacob.montgomery,garnett@wustl.edu`

## Abstract

Choice-based conjoint analysis is an essential tool for learning the marginal effects of multidimensional explanatory features on preferences. However, existing marginal effect models rely either on non-parametric estimators that generalize poorly to individualized effects or on linear latent utility models that completely ignore possible higher-order interactions. We introduce Gaussian Process Conjoint Analysis (GPCA) for learning marginal effects from observed choices as the first-order derivatives of unknown systems. Additionally, we propose a Gaussian mixture approximation for the predictive distributions of marginal effects, which facilitates downstream tasks such as adaptive experimentation. Our experiments show that GPCA achieves more precise and efficient estimation of marginal effects.

## 1 Introduction

Understanding the relationship between targeted outcomes and features in survey experiments is fundamental in many disciplines, including social science [1–3], human-computer interaction [4, 5], and marketing research [6–8]. These associations are often captured by marginal effects, defined as the change in predicted outcomes resulting from changes in features. Depending on the type of attributes, marginal effects can either be computed as the discrete change in outcomes for categorical attributes or as infinitesimal margins for continuous attributes. In survey experiments, marginal effects are often estimated using choice-based conjoint experiments, which present a series of profile pairs with varying attribute values to compare differences in averaged outcomes [6]. For example, researchers might alternate background characteristics to study bias toward immigrants, or system designers might adjust interface setups to improve click-through rates for a new web interface. However, the multidimensional nature of conjoint experiments makes the estimation of heterogeneous marginal effects challenging and not scalable due to interactions with other attributes and small-sample biases. Existing estimators either rely on stacking multiple attributes in a difference-in-differences style [1, 2] or on linear utility theory, which overlooks possible feature expansions [8–11].

We propose Gaussian Process Conjoint Analysis (GPCA), which automatically learns higher-order interactions within the preference learning framework. Marginal effects are derived using the first-order derivatives of a Gaussian Process trained on observed preferences, and their distributions are approximated via Gaussian mixture models. Additionally, we leverage adaptive experimentation for data collection, balancing between exploiting attributes that are more crucial to preferences and exploring attributes where the model exhibits uncertainty, guided by a predictive belief model of the system. As demonstrated in simulated experiments, GPCA achieves more precise estimation of marginal effects compared to other non-parametric and parametric methods.

Workshop on Bayesian Decision-making and Uncertainty, 38th Conference on Neural Information Processing Systems (NeurIPS 2024).

## 2 Related work

**Marginal effects** are a common quantitative method for understanding transformed features in regression models [12] or examining heterogeneous associations between features and outcomes [13]. For instance, marginal effects have been extensively studied in economics to measure the responsiveness of economic variables through the concept of elasticity [14], which quantifies how a percentage change in price corresponds to a percentage change in demand quantity. Marginal effects have also been utilized to enhance the interpretability of machine learning models. Silva Filho et al. [15] proposed a feature importance method for interpreting classification models based on marginal local effects. Merz et al. [16] introduced a marginal attribution method that conditions on quantiles to analyze global gradients in deep neural networks. Scholbeck et al. [17] presented forward marginal effects as a unified, mixed-type feature interpretation method for non-linear machine learning models.

**Conjoint analysis** is an experimental design used to elicit user preferences in recommendation systems [10, 18] and has been widely adopted in quantitative research to learn marginal effects by randomizing over attribute profiles [1, 2]. Existing estimators are predominantly non-parametric and primarily focus on discrete attributes. Hainmueller et al. [1] proposed a difference-in-differences interaction effect estimator to infer preferences from multidimensional choices in survey experiments. Similarly, Egami and Imai [2] introduced a new effect estimator for factorial experiments that does not rely on baseline conditions and generalizes effectively to higher-order interaction effects. In contrast, conjoint analysis involving continuous attributes often employs parametric latent utility functions, such as generalized logistic regression [10], support vector machines [9, 8, 11], Gaussian Processes [19, 18, 20–22], and decision trees [17]. However, these methods are primarily designed for profile recommendation rather than marginal effect estimation.

**Adaptive experimentation** leverages previously collected responses to inform experiment design or data acquisition in subsequent iterations, aiming to maximize the utility of limited data. This approach has been widely adopted by domain scientists to accelerate scientific discovery. For example, Bayesian optimization through adaptive sample selection has been successfully applied in materials science for discovering new materials [23] and in clinical trials for determining the maximum tolerated dose [24, 25]. Similarly, active search has been utilized for the iterative design of virtual screening trials in chemoinformatics [26]. In machine learning, Chen et al. [27] explored the pairwise ranking problem in a crowdsourcing setup using online learning. Bıyık et al. [28] proposed an active, preference-based learning method based on information gain to optimize reward functions in robotics. However, prior adaptive designs in quantitative research have predominantly focused on treatment selection in bandit settings [29–31], with limited attention to marginal effect estimation, particularly within the GP preference learning framework.

## 3 Problem statement

We describe notations and define marginal effects in the setup of Gaussian process conjoint analysis, which are computed as the gradients of the preference probabilities w.r.t the profiles.

**Notations.** Formally, let $\mathbf{x} \in \mathbb{R}^d$ represent the full profile of $d$-dimensional attributes, where $\mathbf{x}l$ denotes the $l$th attribute and $\mathbf{x}-l$ represents all attributes except the $l$th. For pairwise comparisons, let $y_{ij} \in 0, 1$ indicate whether the left-side profile $\mathbf{x}^{(i)}$ is preferred over the right-side profile $\mathbf{x}^{(j)}$, where $y_{ij} = 1$ if $\mathbf{x}^{(i)} \succ \mathbf{x}^{(j)}$ and $y_{ij} = 0$ otherwise. We focus on choice-based conjoint analysis with pairwise comparisons, as scenarios involving multiple choices can be decomposed into multiple pairwise comparisons. For example, if $\mathbf{x}^{(i)}$ is the most preferred option among $\{\mathbf{x}^{(i)}, \mathbf{x}^{(j)}, \mathbf{x}^{(k)}\}$, this can be expressed as $\mathbf{x}^{(i)} \succ \mathbf{x}^{(j)}$ and $\mathbf{x}^{(i)} \succ \mathbf{x}^{(k)}$. Similarly, our notation accommodates score-based conjoint experiments, where $\mathbf{x}^{(i)} \succ \mathbf{x}^{(j)}$ may indicate that $\mathbf{x}^{(i)}$ has a higher score than $\mathbf{x}^{(j)}$. Finally, suppose all observed preferences are collected into the dataset $\mathcal{D} = \{(\mathbf{x}^{(i)}, \mathbf{x}^{(j)}), y_{ij}\}$.

**Gaussian process conjoint analysis.** Conjoint analysis can be formulated as a preference learning problem involving a latent utility function $u(\mathbf{x})$. The preferential relation between $\mathbf{x}^{(i)}$ and $\mathbf{x}^{(j)}$ is determined by comparing their utilities $u(\mathbf{x}^{(i)})$ and $u(\mathbf{x}^{(j)})$. Using a sigmoid probabilistic model $\sigma(\cdot)$, the probability of observed the preference $p(\mathbf{x}^{(i)} \succ \mathbf{x}^{(j)})$ is given by $\sigma\big(u(\mathbf{x}^{(i)}) - u(\mathbf{x}^{(j)})\big)$, allowing for potential labeling errors. Gaussian process conjoint analysis (GPCA) assumes a GP prior on the latent utility $u(\mathbf{x}) \sim \mathcal{GP}(0, K)$, where $K(x, x') = \exp(-\|x - x'\|^2/2)$ is

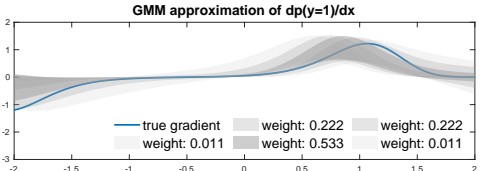
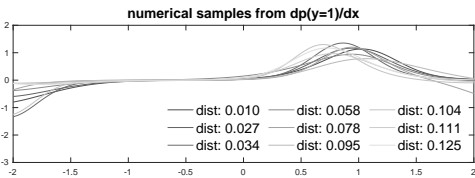

Figure 1: Visualization of the proposed GMM for approximating one-side marginal effect. Left figure shows our GMM approximation of the one-side marginal effect using 5 sampling points, and right figure shows 9 possible true effects obtained by numerical sampling. Darker colors indicate components with higher weights in the GMM and numerical samples closer to the one-side marginal effect posterior mode.

an RBF kernel. The observation model uses a cumulative standard normal function, with $p\big(\mathbf{x}^{(i)} \succ \mathbf{x}^{(j)} \mid u(\mathbf{x}^{(i)}), u(\mathbf{x}^{(j)})\big) = \Phi\big(u(\mathbf{x}^{(i)}) - u(\mathbf{x}^{(j)})\big)$. Although the posterior of $u(\mathbf{x})$ is not analytical in this classification setting, it can be approximated using standard methods such as Laplace approximation or expectation propagation [32, 18]. In some cases, the predictive probability is defined directly on pairs of profiles $(\mathbf{x}^{(i)}, \mathbf{x}^{(j)})$ using the preference kernel $K_{\text{pref}}\big((\mathbf{x}_1^{(i)}, \mathbf{x}_1^{(j)}), (\mathbf{x}_2^{(i)}, \mathbf{x}_2^{(j)})\big) = K(\mathbf{x}_1^{(i)}, \mathbf{x}_2^{(i)}) - K(\mathbf{x}_1^{(i)}, \mathbf{x}_2^{(j)}) - K(\mathbf{x}_2^{(i)}, \mathbf{x}_1^{(j)}) + K(\mathbf{x}_1^{(j)}, \mathbf{x}_2^{(j)})$. We adopted this preference kernel in our implementation of GPCA.

**Marginal effects in GPCA.** We consider the *marginal* effects of profile pairs $(\mathbf{x}^{(i)}, \mathbf{x}^{(j)})$ as the first-order gradients of the preference probability w.r.t both profiles. Leveraging the affine property of Gaussian processes, the gradient $\pi\big((\mathbf{x}^{(i)}, \mathbf{x}^{(j)})\big)$ of the probability of target profile $\mathbf{x}^{(i)}$ being preferred to the opponent profile $\mathbf{x}^{(j)}$ is (see Appendix for full mathematical details)

$$\pi\big((\mathbf{x}^{(i)}, \mathbf{x}^{(j)})\big) = \mathbb{E}_{u|\mathcal{D}}\Big[\phi\big(u(\mathbf{x}^{(i)}) - u(\mathbf{x}^{(j)})\big)\big(\nabla u(\mathbf{x}^{(i)}), -\nabla u(\mathbf{x}^{(j)})\big)\Big] \qquad (1)$$

Intuitively, marginal effects in the outcome space are computed as the expected gradient $\big(\nabla u(\mathbf{x}^{(i)}), -\nabla u(\mathbf{x}^{(j)})\big)$ in the latent utility space, weighted by the densities $\phi\big(u(\mathbf{x}^{(i)}) - u(\mathbf{x}^{(j)})\big)$ of a normal distribution at the latent utility difference $u(\mathbf{x}^{(i)}) - u(\mathbf{x}^{(j)})$. When projected along any unit vector $\hat{\mathbf{e}}_l$, the averaged attribute-specific effect for attribute $l$ on preferences over the profile distribution $\mathcal{P}$ can be further expressed as $\pi_l(\mathbf{x}_l^{(i)}) = \sum_{(\mathbf{x}_{-l}^{(i)}, \mathbf{x}^{(j)}) \sim \mathcal{P}} \langle \pi\big((\mathbf{x}^{(i)}, \mathbf{x}^{(j)})\big), \hat{\mathbf{e}}_l \rangle$.

## 4 Our approach: GMM estimator and adaptive experimentation

In this section, we describe two key elements of our approach: 1) a Gaussian Mixture Model (GMM) approximation to compute $\pi\big((\mathbf{x}^{(i)}, \mathbf{x}^{(j)})\big)$, and 2) an adaptive experimentation strategy that efficiently selects profile pairs for comparison based on uncertainty in the latent utility or preference prediction.

**Gaussian mixture approximation of marginal effects.** Since $\pi\big((\mathbf{x}^{(i)}, \mathbf{x}^{(j)})\big)$ involves taking weighted averages of $\nabla u(\mathbf{x})$ over $u(\cdot)$, we propose the use of Gaussian mixture model (GMM) for approximation. Each component of the GMM is formed by scaling the multivariate Gaussian with the transformed values of quadrature points of the univariate Gaussian determined by Gauss-Hermite quadrature. Formally, let $N$ be the number of points in the quadrature, $k_r$ be the roots of the physicists' version of the Hermite polynomial $H_N(k)$ and $\omega_r = \frac{2^{N-1}N!}{N^2[H_{N-1}(k_r)]^2}$ be the weights of each component [33]. Notice that the gradient of a differentiable GP is itself a GP, so one can first derive a joint posterior of utility function $u(\cdot)$ and its gradient $\nabla u(\mathbf{x})$ from the induced prior $\begin{bmatrix} u \\ \nabla u \end{bmatrix} \sim$

$\mathcal{GP}\big( \begin{bmatrix} \mu_u \\ \nabla \mu_u \end{bmatrix}, \begin{bmatrix} K_u & \nabla K_u^T \\ \nabla K_u & \nabla^2 K_u \end{bmatrix} \big)$ by conditioning on the preference observations. Under this joint

posterior, $\pi\big((\mathbf{x}, \mathbf{x}^{(j)})\big)$ can then be approximated as $\sum_{r=1}^{N} \omega_r \phi\big(\bar{f}_r(\mathbf{x})\big) \circ \mathcal{N}\big(\nabla \mu_u(\mathbf{x}), \nabla^2 K_u(\mathbf{x}, \mathbf{x})\big)$.

Here $\bar{f}_r(\mathbf{x}) = \sqrt{2}[\sigma_u^2(\mathbf{x}) + \sigma_u^2(\mathbf{x}^{(j)})]^{1/2} k_r + [\mu_u(\mathbf{x}) - \mu_u(\mathbf{x}^{(j)})]$ are locations of mixture components defined on the sample point $k_r$s, and $\circ$ denotes the Hadamard (element-wise) product. Figure 1 shows the visualization of the proposed GMM for approximating one-side marginal effect. The left-hand side shows our GMM approximation of the one-sided marginal effect using 5 sampling points, and the

right-hand side shows 9 possible true effects obtained by numerical sampling. Darker colors indicate components with higher weights in the GMM and numerical samples closer to the one-side marginal effect posterior mode. We found $N = 10$ quadrature points sufficient for our GMM.

**Adaptive experimentation in GPCA.** Informed by the posterior belief on the latent utility, adaptive experimentation may efficiently explore attributes whose marginal effects on preferences are less certain. Hence, we determine the next pairs of profiles to compare by maximizing an *acquisition function* $(\mathbf{x}_*^{(i)}, \mathbf{x}_*^{(j)}) = \max_{(\mathbf{x}^{(i)}, \mathbf{x}^{(j)}) \sim \mathbb{P}} \alpha\big((\mathbf{x}^{(i)}, \mathbf{x}^{(j)}); \mathcal{D}\big)$. For simplicity, let $A = u(\mathbf{x}^{(i)}) - u(\mathbf{x}^{(j)})$ and $B = K_{u|\mathcal{D}}(\mathbf{x}^{(i)}, \mathbf{x}^{(i)}) + K_{u|\mathcal{D}}(\mathbf{x}^{(j)}, \mathbf{x}^{(j)})$. We consider the following policies: (1) *Upper confident bound* (UCB) maximizes the 95% confidence interval of preference prediction: $\alpha\big((\mathbf{x}^{(i)}, \mathbf{x}^{(j)}); \mathcal{D}\big) = \big| A + 1.96\sqrt{B} \big|$; (2) *Differential entropy of the latent utility* (DE-U) maximizes the log variance of utility posterior: $\alpha\big((\mathbf{x}^{(i)}, \mathbf{x}^{(j)}); \mathcal{D}\big) = \frac{1}{2}\log(2\pi B) + \frac{1}{2}$; (3) *Differential entropy of the marginal effects* (DE-ME) maximizes their log variance: $\alpha\big((\mathbf{x}^{(i)}, \mathbf{x}^{(j)}); \mathcal{D}\big) =$
$$\log \Big| \sum_{k \in \{i,j\}} \sum_{r=1}^{N} \omega_r \phi\big(\bar{f}_r(\mathbf{x}^{(k)})\big) \phi\big(\bar{f}_r(\mathbf{x}^{(k)})\big)^T \circ \nabla^2 K_{u|\mathcal{D}}(\mathbf{x}^{(k)}, \mathbf{x}^{(k)}) \Big|;$$ (4) *Bayesian active learning by disagreement* (BALD) maximizes the mutual information between utility and predictive preferences: $\alpha\big((\mathbf{x}^{(i)}, \mathbf{x}^{(j)}); \mathcal{D}\big) = \mathbf{I}(y_{ij}, u; \mathbf{x}^{(i)}, \mathbf{x}^{(j)}, \mathcal{D}) \approx h\Big(\Phi\big(\frac{A}{\sqrt{B+1}}\big)\Big) - \frac{C}{\sqrt{B+C^2}} \exp\big(-\frac{A^2}{2(B+C^2)}\big)$, with entropy function $h(p) = -p\log(p) - (1-p)\log(1-p)$ and constant $C = \sqrt{\pi \log(2)/2}$. While UCB emphasizes *exploiting* current belief to find the most preferred profile, DE-U, DE-ME and BALD focus on *exploring* the profile space by reducing uncertainty on marginal effects and predictive preferences.

# 5 Experimental results

Our evaluation on GPCA are based on synthetic data where the functional relations are known analytically. Experiments of two real-world datasets can be found in appendix.

**Data generating process.** Following the simulation specification in Chu and Ghahramani [18], we consider two datasets with discrete (2DPLANE) and continuous (FRIEDMAN) attributes.[1] The 2DPLANE dataset has 5 discrete attributes where $x_1 \in \{-1, 1\}$ and $x_2, \ldots, x_5 \in \{-1, 0, 1\}$, with a piecewise linear utility $u(\mathbf{x}) = 1 + 2x_2 - x_3$ if $x_1 = -1$ and $u(\mathbf{x}) = 1 + x_4 - 2x_5$ if $x_1 = 1$. The FRIEDMAN dataset has 3 continuous attributes where $x_1, \ldots, x_3 \sim [0, 1]$ with a non-linear utility $u(\mathbf{x}) = 3\sin(\pi x_1 x_2) + 6(x_3 - 0.5)^2$. We randomly sample 1000 pairs of profiles in each dataset and set $y_{ij} = 1$ with probability of $\Phi\big(u(\mathbf{x}^{(i)} - u(\mathbf{x}^{(j)}))\big)$ and $y_{ij} = 0$ otherwise. In adaptive experimentation, we initialize with the same 25 profile pairs, and updates model posterior every iteration. Both effects are estimated w.r.t the same target profile distribution to ensure comparability.

**Evaluation metrics and baselines.** We consider three metrics: (1) the RMSE of the estimated effects, (2) the correlation (COR) between the estimated effects and true effects, and (3) the log likelihood (LL) of the estimated effects. We also compare our proposed GMM approximation for marginal effects in GPCA to several baselines: (1) the non-parametric diff-in-mean estimator (DIM) [1], where the continuous attributes in FRIEDMAN are first discretized by splitting into equally-spanned intervals, (2) the standard preference learning method with linear utility (LM-GMM) [9, 10, 8, 11], and (3) an ablated GPCA method (GP-MAP) but with MAP estimation of marginal effects.

Table 1: Averaged performance from GP-GMM estimator and baselines on two synthetic datasets.

| DATASET | ESTIMATOR | Marginal effects | | | Component effects | | |
|---|---|---|---|---|---|---|---|
| | | RMSE $\downarrow$ | COR $\uparrow$ | LL $\uparrow$ | RMSE $\downarrow$ | COR $\uparrow$ | LL $\uparrow$ |
| 2DPLANE | DIM | $0.712 \pm 0.022$ | $0.013 \pm 0.003$ | $-2.137 \pm 0.115$ | $0.109 \pm 0.005$ | $0.341 \pm 0.029$ | $0.494 \pm 0.117$ |
| | LM-GMM | $0.213 \pm 0.001$ | $0.340 \pm 0.005$ | $-0.238 \pm 0.145$ | $0.069 \pm 0.002$ | $0.475 \pm 0.019$ | $-0.778 \pm 0.157$ |
| | GP-MAP | $0.175 \pm 0.002$ | $0.732 \pm 0.007$ | $-3.893 \pm 0.863$ | $0.052 \pm 0.002$ | $0.611 \pm 0.024$ | $1.401 \pm 0.177$ |
| | GP-GMM | $\mathbf{0.135 \pm 0.002}$ | $\mathbf{0.803 \pm 0.007}$ | $\mathbf{0.563 \pm 0.023}$ | $\mathbf{0.044 \pm 0.001}$ | $\mathbf{0.616 \pm 0.025}$ | $\mathbf{2.000 \pm 0.082}$ |
| FRIEDMAN | DIM | $0.910 \pm 0.008$ | $0.024 \pm 0.005$ | $-9.658 \pm 0.392$ | $0.150 \pm 0.010$ | $0.944 \pm 0.017$ | $-1.824 \pm 0.480$ |
| | LM-GMM | $0.845 \pm 0.010$ | $0.328 \pm 0.007$ | $-1.001 \pm 0.271$ | $0.078 \pm 0.005$ | $0.980 \pm 0.005$ | $0.503 \pm 0.245$ |
| | GP-MAP | $0.510 \pm 0.008$ | $0.830 \pm 0.006$ | $-3.869 \pm 0.530$ | $\mathbf{0.042 \pm 0.003}$ | $\mathbf{0.995 \pm 0.001}$ | $1.680 \pm 0.045$ |
| | GP-GMM | $\mathbf{0.478 \pm 0.008}$ | $\mathbf{0.847 \pm 0.005}$ | $\mathbf{-0.213 \pm 0.065}$ | $\mathbf{0.042 \pm 0.003}$ | $\mathbf{0.995 \pm 0.001}$ | $\mathbf{1.689 \pm 0.044}$ |

---

[1]See https://www.dcc.fc.up.pt/~ltorgo/Regression/DataSets.html for details.

We then investigate adaptive experimentation in GPCA for increasing efficiency of effect estimation. We consider various policies: (1) UCB popular in multi-arm bandit setting [34], (2) DE-U and DE-ME for active learning based on differential entropy [35–37], (3) BALD in Bayesian active learning for model uncertainty reduction [38] and (4) UNIFORM design in non-parametric conjoint analysis [1, 2].

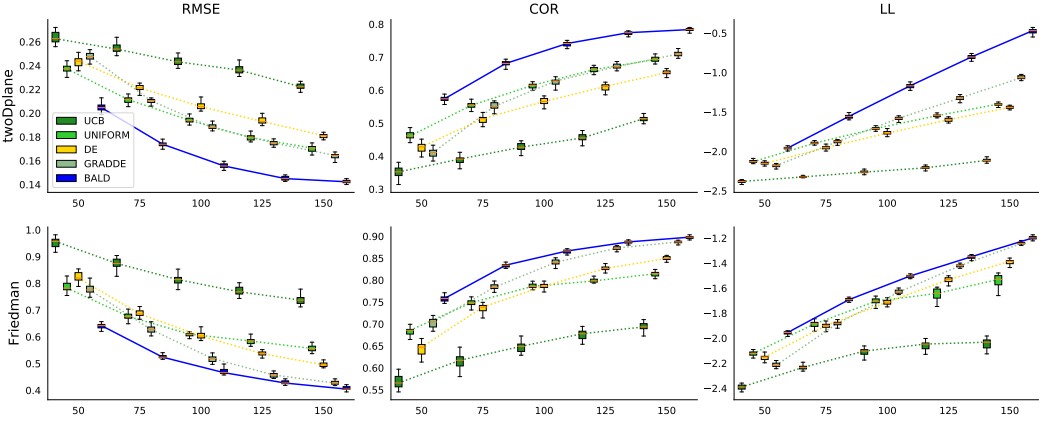

Figure 2: Box plots of averaged RMSE, COR and LL of marginal effects under different acquisition policies.

**Results.** Table 1 shows the averaged RMSE, correlation and log likelihood across 25 different seeds of the proposed GP-GMM and baselines, where GP-GMM consistently has more precise effect estimation with lower RMSE and higher COR/LL for both effects. Figure 2 shows box plots of averaged RMSE, COR and LL of marginal effect estimation with adaptive experimentation under different acquisition policies. Sample size range from 50 to 150, and performance metrics are reported every other 25 acquisitions. Overall BALD (blue) outperforms the rest of policies including UNIFORM and UCB, indicating higher efficiency for effect estimation when the acquisition is designed to reduce model uncertainty. Morever, UCB (forest green) has overall the worst performance in estimating both marginal and component effects as it solely reinforces current belief on the probability of preference. Results for component effect estimation can be found in Appendix.

Besides estimation of marginal effects, we also examine the model quality of GPCA by evaluating the prediction accuracy of unrevealed preferences among the not acquired profile pairs. Figure 3 shows the averaged accuracy and STDs of preference prediction by various policies. With as few as 50 data points, GPCA manages to predict at least 80% of the unrevealed preference and 95% when 150 data points are adaptively acquired by BALD.

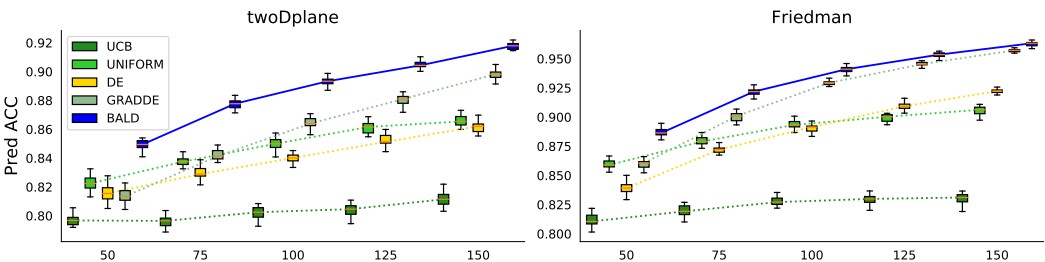

Figure 3: Averaged accuracy and STDs of preference prediction by various policies for simulated data.

## 6 Conclusion

We introduce GPCA, a Gaussian Process conjoint analysis model for estimating marginal effects in choice-based conjoint experiments with a Gaussian mixture approximation for their distributions that could enhance precision and efficiency in effect estimation aided by adaptive experimentation. Experiments show that GPCA achieves more precise and efficient estimation of marginal effects, suggesting great potential in either machine learning interpretation or industrial applications.

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

# A    DIM: marginal effects of discrete attributes

In conjoint analysis with factorial design, attributes usually take discrete values of different levels $\mathbf{x}_l = 1, \ldots, C_l$. For a target distribution of profiles $\mathcal{P}$, the marginal effects $\pi_l(a, b)$ of attribute $\mathbf{x}_l$ from level $a$ to $b$ ($1 \leq a < b \leq C_l$) are captured by the average marginal component effect (AMCE), defined as the difference in expected preferential outcomes averaged over all the possible values of the remaining attributes $\mathbf{x}_{-l}$ over $\mathcal{P}$:

$$\pi_l(a, b) = \mathbb{E}_{\mathbf{x}_{-l}^{(i)}, \mathbf{x}^{(j)} \sim \mathcal{P}}[y_{ij} \mid \mathbf{x}_l^{(i)} = b] - \mathbb{E}_{\mathbf{x}_{-l}^{(i)}, \mathbf{x}^{(j)} \sim \mathcal{P}}[y_{ij} \mid \mathbf{x}_l^{(i)} = a] \tag{2}$$

Intuitively, $\pi_l(a, b)$ represents the increase in the probability of one profile being preferred if the $l$th attribute were $b$ instead of $a$ for profile distribution $\mathcal{P}$. With the conditionally independent assumption, $\pi_l(a, b)$ can be estimated straight-forwardly using a difference-in-mean approach:

$$\hat{\pi}_l(a, b) = \frac{\sum\limits_{(\mathbf{x}^{(i)}, \mathbf{x}^{(j)}) \in \mathcal{D}} y_{ij} \mathbb{I}[\mathbf{x}_l^{(i)} = b]}{\sum\limits_{(\mathbf{x}^{(i)}, \mathbf{x}^{(j)}) \in \mathcal{D}} \mathbb{I}[\mathbf{x}_l^{(i)} = b]} - \frac{\sum\limits_{(\mathbf{x}^{(i)}, \mathbf{x}^{(j)}) \in \mathcal{D}} y_{ij} \mathbb{I}[\mathbf{x}_l^{(i)} = a]}{\sum\limits_{(\mathbf{x}^{(i)}, \mathbf{x}^{(j)}) \in \mathcal{D}} \mathbb{I}[\mathbf{x}_l^{(i)} = a]} \tag{3}$$

However, this difference-in-mean approach for estimating marginal effects suffers from two issues. First, generalization of this estimator for heterogeneous effect resulting from either background characteristics or high-level interactions could get more complicated as calculation of multiple differences is required. For instance, for obtaining interaction effects of between $\mathbf{x}_l^{(i)}$ and $\mathbf{x}_m^{(i)}$ from level $c$ to $d$ in $\mathbf{x}_m^{(i)}$, one needs to compute $[\hat{\pi}_l(a, b)\big|_{\mathbf{x}_m^{(i)} = c} - \hat{\pi}_l(a, b)\big|_{\mathbf{x}_m^{(i)} = c}] - [\hat{\pi}_l(a, b)\big|_{\mathbf{x}_m^{(i)} = d} - \hat{\pi}_l(a, b)\big|_{\mathbf{x}_m^{(i)} = d}]$ [1]. Second, in practice, continuous attributes are rarely repeated and thus often need to be discretized into multiple levels; otherwise, each level $\mathbf{x}_l^{(i)} = a$ would have very few observations. However, this discretization is subject to the chosen cutoff points and may lead to an oversimplification of the system, threatening the internal validity of marginal effect estimation.

# B    Mathematical details of GPCA

## B.1    Predictive distribution of Gaussian process preference learning

The inferred latent utility posterior could also be used for prediction. For any new pair of profiles $(\mathbf{x}_*^{(i)}, \mathbf{x}_*^{(j)})$, suppose their corresponding utility vector has been approximated by a bivariate normal $\mathbf{u}_* = [u(\mathbf{x}_*^{(i)}), u(\mathbf{x}_*^{(j)})]^T \sim \mathcal{N}(\boldsymbol{\mu}_*, \boldsymbol{\Sigma}_*)$. Let $\boldsymbol{\mu}_* = [\mu_*^{(i)}, \mu_*^{(j)}]^T$ and $\sigma_*^2 = 1 + [1, -1]\boldsymbol{\Sigma}_*[1, -1]^T$, then the predictive probability has the following closed-form:

$$p(\mathbf{x}_*^{(i)} \succ \mathbf{x}_*^{(j)}) = \int \Phi\big(u(\mathbf{x}_*^{(i)}) - u(\mathbf{x}_*^{(j)})\big) p(\mathbf{u} \mid \mathcal{D}) d\mathbf{u} = \Phi\big(\frac{\mu_*^{(i)} - \mu_*^{(j)}}{\sigma_*}\big) \tag{4}$$

## B.2    Marginal effects and Gaussian mixture approximation

By exploiting the affine property of Gaussian processes, the gradient $\pi\big((\mathbf{x}^{(i)}, \mathbf{x}^{(j)})\big)$ in probability of target profile $\mathbf{x}^{(i)}$ being preferred to opponent profile $\mathbf{x}^{(j)}$ can be derived as:

$$\pi\big((\mathbf{x}^{(i)}, \mathbf{x}^{(j)})\big) = \frac{\partial}{\partial(\mathbf{x}^{(i)}, \mathbf{x}^{(j)})}[p(\mathbf{x}^{(i)} \succ \mathbf{x}^{(j)})] \qquad \text{definition of AMCE} \tag{5}$$

$$= \frac{\partial}{\partial(\mathbf{x}^{(i)}, \mathbf{x}^{(j)})} \mathbb{E}_{u \mid \mathcal{D}}\Big[\Phi\big(u(\mathbf{x}^{(i)}) - u(\mathbf{x}^{(j)})\big)\Big] \qquad \text{averaged by } u \mid \mathcal{D} \tag{6}$$

$$= \mathbb{E}_{u \mid \mathcal{D}}\Big[\phi\big(u(\mathbf{x}^{(i)}) - u(\mathbf{x}^{(j)})\big)\big(\nabla u(\mathbf{x}^{(i)}), -\nabla u(\mathbf{x}^{(j)})\big)\Big] \quad \text{chain rule} \tag{7}$$

Note that in the second step we swapped the order of expectation and differentiation. As the gradient of a GP is still a GP, we can first write the joint distribution of utility $u(\cdot) \mid \mathcal{D}$ and utility gradient $\nabla u \mid \mathcal{D}$ under utility posterior $\mathcal{GP}\big(\mu_{u \mid \mathcal{D}}(\mathbf{x}), K_{u \mid \mathcal{D}}(\mathbf{x}, \mathbf{x}')\big)$ on $\mathcal{D}$ as:

$$\begin{bmatrix} u \mid \mathcal{D} \\ \nabla u \mid \mathcal{D} \end{bmatrix} \sim \mathcal{GP}\big( \begin{bmatrix} \mu_{u \mid \mathcal{D}} \\ \nabla \mu_{u \mid \mathcal{D}} \end{bmatrix}, \begin{bmatrix} K_{u \mid \mathcal{D}} & \nabla K_{u \mid \mathcal{D}}^T \\ \nabla K_{u \mid \mathcal{D}} & \nabla^2 K_{u \mid \mathcal{D}} \end{bmatrix} \big) \tag{8}$$

where $\nabla \mu_{u|\mathcal{D}} = \partial \mu_{u|\mathcal{D}}(\mathbf{x})/\partial \mathbf{x}$ is the first-order derivative of the posterior mean, $\nabla K_{u|\mathcal{D}} = \partial K_{u|\mathcal{D}}(\mathbf{x}, \mathbf{x}')/\partial \mathbf{x}$ is the first-order derivative of the posterior covariance and $\nabla^2 K_{u|\mathcal{D}} = \partial^2 K_{u|\mathcal{D}}(\mathbf{x}, \mathbf{x}')/\partial \mathbf{x} \partial \mathbf{x}'$ is its second-order mixed derivatives. We use a Gaussian mixture model (GMM) to approximate $\mathbf{g}(\mathbf{x}; \mathbf{x}^{(j)}, \mathcal{D})$. Each component of the GMM is formed by scaling the multivariate Gaussian with the transformed values of quadrature points of the univariate Gaussian determined by Gauss-Hermite quadrature. Let $N$ be the number of points in the quadrature, $k_r$ be the roots of the physicists' version of the Hermite polynomial $H_N(k)$ and $\omega_r = \frac{2^{N-1} N!}{N^2 [H_{N-1}(k_r)]^2}$ be the weights of each component [33]. We could then approximate $\mathbf{g}(\mathbf{x}; \mathbf{x}^{(j)}, \mathcal{D})$ as:

$$\mathbf{g}(\mathbf{x}; \mathbf{x}^{(j)}, \mathcal{D}) \approx \sum_{r=1}^{N} \omega_r \phi\big(\bar{f}_r(\mathbf{x})\big) \circ \mathcal{N}\Big(\nabla \mu_{u|\mathcal{D}}(\mathbf{x}), \nabla^2 K_{u|\mathcal{D}}(\mathbf{x}, \mathbf{x})\Big) \tag{9}$$

$$= \sum_{r=1}^{N} \omega_r \mathcal{N}\Big(\phi\big(\bar{f}_r(\mathbf{x})\big) \circ \nabla \mu_{u|\mathcal{D}}(\mathbf{x}), \phi\big(\bar{f}_r(\mathbf{x})\big)\phi\big(\bar{f}_r(\mathbf{x})\big)^T \circ \nabla^2 K_{u|\mathcal{D}}(\mathbf{x}, \mathbf{x})\Big) \tag{10}$$

where $\bar{f}_r(\mathbf{x}) = \sqrt{2}[\sigma^2_{u|\mathcal{D}}(\mathbf{x}) + \sigma^2_{u|\mathcal{D}}(\mathbf{x}^{(j)})]^{1/2} k_r + [\mu_{u|\mathcal{D}}(\mathbf{x}) - \mu_{u|\mathcal{D}}(\mathbf{x}^{(j)})]$ are locations of mixture components defined on the sample point $k_r$s, and $\circ$ denotes the Hadamard (element-wise) product.

## C   Additional experiment results

Table 2 shows the averaged accuracy and STDs of preference prediction from GPCA and baselines on both synthetic datasets. GPCA has the best prediction for capturing the underlying preferential relations in the system.

Table 2: Averaged accuracy and STDs of preference prediction from GPCA and baselines on both synthetic datasets. GPCA has the best prediction for capturing the underlying preferential relations in the system.

| DATASET | 2DPLANE | | | FRIEDMAN | | |
|---|---|---|---|---|---|---|
| | DIM | SVM | GPCA | DIM | SVM | GPCA |
| ACC | 0.696±0.006 | 0.824±0.003 | **0.986±0.002** | 0.785±0.006 | 0.795±0.005 | **0.956±0.002** |

Figure 4 shows performance of different acquisition policies for component effect estimation in the two synthetic data.

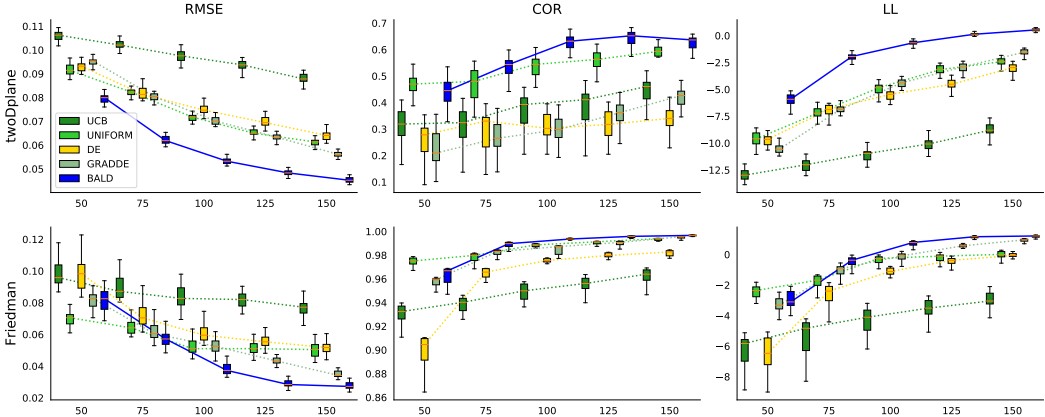

Figure 4: Box plots of averaged RMSE, COR and LL and their STDs of component effects with adaptive experimentation.

# D   Experimental results from two real-world applications

**Data.** We apply GPCA to two real-world conjoint experiments: U.S. citizens' preferences across presidential candidates and attitudes toward immigrants containing 1733 and 6980 pairwise comparisons [1, 39]. Attributes in the candidate experiment include various aspects of candidates' personal background, demographics and issue positions, such as religion, education, profession, income and race, while attributes in the immigrant experiment include employment plans, job experience, language skills, country of origin, reasons for applying and so on.

Table 3: List of attributes with estimated component effects by GPCA and DIM used in the original studies, grouped by negative, null and positive effects.

| DATASET | GPCA \ DIM | NEG | NULL | POS |
|---|---|---|---|---|
| Candidate | NEG | Evangelical protestant, Mormon, car dealer,Age 68 | Jewish,Catholic,high school teacher, farmer,Income 210K,Black,Age 60 | — |
| | NULL | — | Mainline protestant,Lawyer,doctor,female, Income 54K,Hispanic,Asian American,Age 52 | Baptist college,Income 65K |
| | POS | — | Income 92K,5.1M,Caucasian, Native American,Age 45,75 | Military,community college, state university,Ivy League |
| Immigrant | NEG | India,China,will look for work, interview with employer,once as tourist | Broken English,Used interpreter,Germany, France,Mexico,Philippines,Poland,Iraq | — |
| | NULL | — | Mainline protestant,Lawyer,doctor,female, Income 54K,Hispanic,Asian American,Age 52 | — |
| | POS | — | Male,Somalia,financial analyst, waiter,child care provider | college degree,graduate degree,teacher,nurse,doctor computer programmer,research scientist,escape persecution |

**Results.** We run GPCA using all samples in both datasets. Table 3 shows the list of attributes with estimated component effects by GPCA and DIM used in original studies grouped by negative, null and positive effects. Overall, component effect estimation by GPCA is more reasonable. For example, in the candidate experiment GPCA found negative effects of Black candidates working as high school teachers or farmers on the probability of becoming U.S. presidents and positive effects of Caucasian candidates with 5.1M or more annual income, while DIM found no effects for any of these attributes. In the immigrant experiment, GPCA found negative effects of Iraqi applicants with broken English on the probability of immigration approval and positive effects of applicants working as financial analysts, while DIM found no effects. Figure 5 shows the averaged accuracy and STDs of preference prediction by various policies for real data with sample size varying from 100 to 800, where BALD has better prediction of unrevealed preferences than randomized policy.

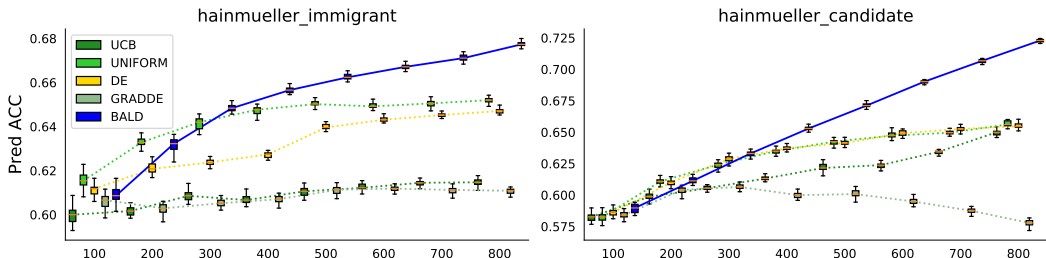

Figure 5: Averaged accuracy and STDs of preference prediction by various policies for real data, with sample size varying from 100 to 800. BALD has better prediction of unrevealed preferences than randomized policy.

