# OpenReview forum: "Gaussian Process Conjoint Analysis for Adaptive Marginal Effect Estimation"
_NeurIPS.cc/2024/Workshop/BDU — NeurIPS BDU Workshop 2024 Poster_

### Official Review · Reviewer_vcnk · 2024-09-19
**Peer Review for Gaussian Process Conjoint Analysis for Adaptive Marginal Effect Estimation**

**Rating:** 7
**Confidence:** 3

**Review:**

## 1. Summary
The paper presents Gaussian Process Conjoint Analysis (GPCA), a novel approach for estimating marginal effects in choice-based conjoint experiments. The authors propose a method that derives marginal effects as first-order derivatives from observed choices through Gaussian processes and introduce a Gaussian mixture approximation for predictive distributions of marginal effects. GPCA's experimental results on synthetic data demonstrate a more precise and efficient marginal effect estimation.

Let me mention that I am not an expert on Gaussian process, so it could be beneficial to have another reviewer review and validate the mathematical derivation in details. I do have experience with conjoint analysis and adaptive experimentation methods.

## 2. Strengths and Weaknesses
### a. Originality

The approach, GPCA combining Gaussian processes with adaptive experimentation, is indeed novel (i.e., limited literature on this).

The paper could benefit from a more extensive review of recent literatures/developments, but understand the constraint of paper length.

### b. Quality

The findings of the paper are substantiated by experimental results on synthetic data.

However, there are two feedback for the authors to consider -
1. As table 1 suggests, while GP-GMM results are overarchingly better under 2DPLANE dataset, they are only marginally better than GP-MAP under FRIEDMAN. It could be beneficial for the authors to discuss the hypotheses - is it the nature of the dataset (continuous vs. discrete)?
2. In figure 2, 3, and 4, it seems that each acquisition function was run on a different selection of sample size. For example, BALD did not have a result for sample size <= 50 while the other approach did. Is there a specific consideration for this set-up?

### c. Clarity

The paper is generally well-organized and clearly written. The results are documented by tables and visuals and discussed in details.

One feedback for the authors to consider -
1. Consider moving Table 1 under section 5 experimental results, so the section is organized by marginal and component effect estimation (table 1) and adaptive experimentation (figure 2).

### d. Significance

The results presented in this paper carry significance in industrial applications, such as marketing, social sciences, robotics. The fact that GPCA can handle continuous responses are particularly significant and could use more underscoring in the paper.


## 3. Questions
1. How generalizable is GPCA approach across different use cases (especially for real-world, continuous datasets)?
2. Could the authors address potential limitations of using Gaussian processes, such as computational complexity or sensitivity to hyperparameter tuning?

---

### Official Review · Reviewer_wyfE · 2024-09-27
**Technically solid, where reasons to accept outweigh reasons to reject.**

**Rating:** 6
**Confidence:** 4

**Review:**

The paper explores Choice-based Conjoint Analysis (CBC), a popular market research technique used to understand consumer preferences and how they make trade-offs between product attributes. It is an interesting approach to applying GPs to a practical decision problem.

However, there are some concerns regarding this work:

The text should be reviewed for typos and grammatical errors.
Some sections of the text require clearer explanations. For instance, it is not entirely clear how this work differentiates itself from other GP-related studies in the field (e.g., references 12 and 13). These references also have not been used in the experiment section to demonstrate the proposed method's superiority.
Figure 1: The left-hand plot is not very informative. At the very least, using different colors could help highlight the differences between the weights.
Generally, determining the weights in mixture models is a challenging task. The paper does not sufficiently address this issue and instead relies on previously defined weights, which warrants further consideration.
The function alpha mentioned in line 109 has not been defined earlier in the paper, yet it is used in discussions about policies.

---

### Decision · Program_Chairs · 2024-10-09

Accept (Poster)